# Prevalence of Sleep Disorders, Risk Factors and Sleep Treatment Needs of Adolescents and Young Adult Childhood Cancer Patients in Follow-Up after Treatment

**DOI:** 10.3390/cancers14040926

**Published:** 2022-02-13

**Authors:** Shosha H. M. Peersmann, Martha A. Grootenhuis, Annemieke van Straten, Gerard A. Kerkhof, Wim J. E. Tissing, Floor Abbink, Andrica C. H. de Vries, Jacqueline Loonen, Leontien C. M. Kremer, Gertjan J. L. Kaspers, Raphaële R. L. van Litsenburg

**Affiliations:** 1Princess Máxima Center for Pediatric Oncology, P.O. Box 85090, 3508 AB Utrecht, The Netherlands; s.h.m.peersmann@prinsesmaximacentrum.nl (S.H.M.P.); m.a.grootenhuis-2@prinsesmaximacentrum.nl (M.A.G.); w.j.e.tissing@prinsesmaximacentrum.nl (W.J.E.T.); a.c.h.devries-15@prinsesmaximacentrum.nl (A.C.H.d.V.); l.c.m.kremer@prinsesmaximacentrum.nl (L.C.M.K.); g.j.l.kaspers@prinsesmaximacentrum.nl (G.J.L.K.); 2Emma Children’s Hospital, Amsterdam UMC, Vrije Universiteit, Pediatric Oncology, P.O. Box 7057, 1007 MB Amsterdam, The Netherlands; f.abbink@amsterdamumc.nl; 3Department of Clinical, Neuro and Developmental Psychology, Faculty of Behavioural and Movement Science & Amsterdam Public Health Research Institute, Vrije Universiteit, 1081 BT Amsterdam, The Netherlands; a.van.straten@vu.nl; 4Department of Psychology, University of Amsterdam, P.O. Box 94208, 1090 GE Amsterdam, The Netherlands; info@gerardkerkhof.nl; 5Department Sleep Disorders Center, Haaglanden Medisch Centrum (HMC), 2512 VA The Hague, The Netherlands; 6Beatrix Children’s Hospital, University Medical Center Groningen, University of Groningen, P.O. Box 30.001, 9713 GZ Groningen, The Netherlands; 7Department of Pediatric Hemato-Oncology, Erasmus MC–Sophia Children’s Hospital, P.O. Box 2060, 3015 GD Rotterdam, The Netherlands; 8Department of Hematology, Radboud University Medical Center, P.O. Box 9101, 6500 HB Nijmegen, The Netherlands; jacqueline.loonen@radboudumc.nl; 9University Medical Center Utrecht, Wilhelmina Children’s Hospital, P.O. Box 85090, 3508 AB Utrecht, The Netherlands

**Keywords:** sleep, survivorship care, childhood cancer, adolescents, young adults, adolescent and young adult (AYA), quality of life

## Abstract

**Simple Summary:**

Sleep disorders negatively impact adolescent and young adult childhood cancer patients’ physical and psychosocial health. Early recognition might improve timely treatment. This national cohort study describes subjective sleep disorders (using a validated self-report questionnaire according to international diagnostic criteria) in childhood cancer patients after treatment, including all types of diagnoses. Sleep disorder prevalence rates ranged from 1.8–9.6%. Insomnia and circadian rhythm sleep disorders were most commonly reported and more prevalent than in the general population. Female sex, young adulthood (18–26 years old) and co-morbid health conditions were risk factors for having a sleep disorder, but cancer-related factors were not. The most commonly used sleep medication was melatonin, which exceeded use described in the general population. Patients with a sleep disorder expressed an unmet need for sleep treatment. Screening for sleep disorders after cancer might improve access to treatment and support childhood cancer patients to maintain optimal health and quality of life.

**Abstract:**

Background: Sleep disorders negatively impact adolescent and young adult childhood cancer patients’ physical and psychosocial health. Early recognition improves timely treatment. We therefore studied the prevalence of subjective sleep disorders, risk factors and sleep treatment needs after completion of childhood cancer treatment. Methods: Childhood cancer patients (12–26 years old), ≥6 months after treatment, were invited to fill out the Holland Sleep Disorders Questionnaire, which distinguishes six sleep disorders in substantial agreement with the International Classification of Sleep Disorders, second edition (ICSD-2). They additionally indicated sleep treatment needs. Prevalence rates and needs were displayed in percentages. Logistic regression models were used for risk factors. Results: 576 patients participated (response rate 55.8%)—49.5% females, mean age 17.0 years, 44.4% hemato-oncology, 31.9% solid tumors, 23.6% neuro-oncology. Prevalence rates were: insomnia (9.6%), circadian rhythm sleep disorder (CRSD; 8.1%), restless legs syndrome (7.6%), parasomnia (3.5%), hypersomnia (3.5%) and sleep-related breathing disorders (1.8%). Female sex, comorbid health conditions and young adulthood seem to be risk factors for sleep disorders, but cancer-related factors were not. Differing per sleep disorder, 42–72% wanted help, but only 0–5.6% received sleep treatment. Conclusions: Insomnia and CRSD were most prevalent. An unmet need for sleep treatment was reported by childhood cancer patients during follow-up. Screening for sleep disorders after cancer might improve access to treatment and patient wellbeing.

## 1. Introduction

Sleep problems entail a health burden and are associated with many negative physical (e.g., cardiovascular disease, chronic pain, obesity) and mental health outcomes (e.g., depression, anxiety) and a lower quality of life [1,2,3,4,5]. Adolescents and young adults are at risk for sleep problems [6,7], since this is a developmental period characterized by changes in sleep architecture as well as changed social demands [8,9]. In a longitudinal community-based study following children and adolescents, the presence of a sleep problem increased the chance of reporting a new medical disorder and new medication use in follow-up [1]. Recently, the American Academy of Sleep Medicine published a position paper about the essential role of sleep for health [10] and they stated that the importance of sleep should be more acknowledged within research, clinical care, in hospital sleeping conditions, and in educational programs for health professionals. For example, in a study among children and adolescents in primary care, it was found that sleep disorders were fairly underdiagnosed [11]. This is unfortunate, since most sleep disorders can be successfully treated [12].

A recent position paper in the pediatric oncology field also emphasizes the importance of addressing sleep in research and clinical care [13]. Sleep might be important to monitor in childhood cancer patients during the cancer course; it was associated with relapse in children with acute lymphoblastic leukemia (ALL) [14]. Furthermore, childhood cancer survivors report more sleep impairment than peers [15]. As childhood cancer survivors have an increased risk of adverse physical and mental health outcomes, such as cardiovascular disease, neurocognitive deficits or fatigue [16,17,18], having a sleep disorder on top of this creates a higher risk of health burden.

There are over eighty types of sleep disorders as estimated by the International Classification of Sleep Disorders, second edition (ICSD-2) [19]. The majority can broadly be classified into six main categories: (1) *Insomnias*: these are characterized by difficulty initiating and/or maintaining sleep with a perceived low quality of sleep which significantly impacts daily functioning; (2) *Circadian rhythm sleep disorders (CRSDs*): these include misalignment between the timing of the endogenous sleep–wake rhythm and the desired sleep timing; (3) *Sleep-related breathing disorders (SBDs):* these include disturbed breathing and ventilation during sleep, e.g., sleep apnea; (4) *Hypersomnia disorders*: these entail excessive daytime sleepiness with the inability to stay awake, e.g., narcolepsy; (5) *Parasomnias* are undesirable events occurring during sleep, e.g., night terrors (pavor nocturnus), sleep walking or sleep-related eating; (6) *Restless legs syndromes and limb movements syndromes (RLS/LMSs)* are sleep-related movement disorders. RLS is the strong urge to move one’s legs, which worsens during relaxation and happens mostly in the evening or during the night; LMSs are characterized by repetitive limb movements during sleep.

Childhood cancer patients during and after treatment might be prone to sleep disorders due to the (late) effects of cancer treatments or tumor localization or indirect effects, such as increased psychological distress. For example, during ALL treatment, a disrupted circadian sleep–activity rhythm has been found [20,21]. The treatment influences the circadian rhythm: more fatigue resulting in naps during the day, less physical activity and disruption of the nights due to pain/side effects. This might lead to CRSD. After treatment, patients are also more at risk for obesity [22], which is a risk factor for SBD [13,23,24]. Patients with a brain tumor localized in the area involved in sleep regulation, e.g., the hypothalamus, such as patients with a craniopharyngioma, also report more sleep problems [25,26], such as hypersomnia or SBD [26]. Furthermore, an increased level of psychological distress, which accompanies living with childhood cancer and survivorship [15,16,18], increases the risk of insomnia, a sleep disorder precipitated and perpetuated by distress and hyperarousal [27,28]. Other risk factors might be: more co-morbid health conditions [17,18], which can increase the risk of impaired sleep and insomnia [2,3,4,5,17,18]. Additionally, female sex might be a risk factor [6,11], and differences might also emerge in the developmental age groups of adolescence and young adulthood, such as increased levels of CRSD [7,8,9].

In pediatric oncology, but also in adult oncology, sleep is commonly studied as a general symptom rather than a specific sleep disorder according to diagnostic criteria [13,29]. Prevalence rates for sleep problem symptoms vary in pediatric oncology populations, ranging from 12.1% up to about 87%, depending on treatment phase and follow-up, diagnostic group [30,31,32,33] and depending on the measurement used [34]. Studies of the occurrence of sleep disorders in childhood cancer patients are sparse [35], despite this information being relevant, as underlying sleep disorders to generic sleep complaints are diverse and each requires specific treatment [12]. For example, trouble falling asleep can be caused by insomnia as well as a delayed circadian rhythm, but recommended therapies for these sleep disorders are very distinct. Cognitive behavioral therapy (CBT-i) is recommended for insomnia, whereas chronotherapies, such as light therapy or melatonin to adjust circadian timing, are recommended for circadian rhythm-related disorders [12]. The diversity of sleep disorders in childhood cancer patients was recently illustrated in a case study of childhood cancer patients with a suprasellar tumor. The sleep disturbance of each patient was caused by a different sleep disorder and warranted different treatment strategies, e.g., one patient had a sleep-related breathing disorder for which nocturnal ventilation was used and another suffered from hypersomnia for which modafinil treatment was applied [26]. This highlights the importance of detailed inspection of symptoms to diagnose and treat sleep disorders effectively. It is therefore important to gain more knowledge of the prevalence of and risk factors for specific sleep disorders in childhood cancer patients after completion of treatment.

Furthermore, it is also important to consider the patient’s need for sleep treatment. In the general population, sleep disorders are underdiagnosed [11] and access to sleep treatment is limited due to a shortage of sleep therapists and sleep clinics [10,36]. Information on the prevalence of sleep disorders combined with the patient’s need for sleep treatments will allow for a better organization of (long-term) follow-up care of childhood cancer patients and survivors.

Therefore, the aim of this study is to estimate the prevalence of each of the six main categories of subjective sleep disorders according to diagnostic criteria of the International Classification of Sleep Disorders (ICSD-2) in adolescent and young adult childhood cancer patients during follow-up after treatment. Furthermore, we aimed to assess associated risk factors and describe current sleep treatment needs.

## 2. Materials and Methods

### 2.1. Study Participants

This study is part of the MICADO project (Managing Insomnia after Childhood cancer in ADOlescents), a two-step project aimed at screening for and treatment of sleep problems in pediatric oncology. The results described here were collected during part 1 of the project: a national, observational, cross-sectional screening study. All adolescents and young adults in the Netherlands that were included in the Dutch Childhood Oncology Group (DCOG) and the LATER (Dutch survivors who were at least five years after diagnosis) registry were eligible if they: (1) were aged ≥ 12 years old at the time of study; (2) were diagnosed with cancer or a non-malignant/low-grade tumor for which oncological follow-up was required, before the age of 19 years; (3) were diagnosed within the last 10 years; (4) had finished their last cancer treatment at least 6 months ago, since patients are in the first six months after treatment still physically recovering. Exclusion criteria were: (a) receiving palliative therapy, (b) not being able to fill out questionnaires independently due to a language barrier or cognitive impairment, (c) physician-specific reasons, e.g., difficulties in communication with the family/patients and (d) comorbidities that prohibited effective or safe participation in part 2 of the project, which included an intervention for insomnia: severely diminished eye-sight with no light perception, schizophrenia and severe substance abuse [37].

### 2.2. Procedures

Eligible patients were recruited between November 2018 and July 2021 from five locations: the Princess Máxima Center for pediatric oncology in Utrecht, Amsterdam University Medical Center, University Medical Center Groningen, Radboud University Medical Center Nijmegen, Erasmus Medical Center Rotterdam. The eligibility of each patient and permission to contact them was obtained from the electronic patient files and the treating physician, respectively. Thereafter, patients were approached with an invitation letter to participate and complete the self-report questionnaires described below (paper-and-pencil or online). In the invitation letter, patients with and without sleep problems were explicitly asked to participate. Patients received a 10-euro gift card for participation. After three weeks with no response, a reminder letter including the self-report questionnaires was sent and after six weeks a telephonic reminder took place. Informed consent was obtained from all participants. For patients 12–15 years old, parents also returned an additional informed consent form for their child. The Institutional Review Board of the University Medical Center Utrecht classified this study as exempt from the Medical Research Involving Human Subjects Act (WMO, article 16).

### 2.3. Measures

#### 2.3.1. Holland Sleep Disorder Questionnaire (HSDQ)

The HSDQ was developed by Kerkhof and colleagues (2013) for the screening of sleep disorders in concordance with the criteria of the International Classification of Sleep Disorders (ICSD-2) in adults aged 18–70 years old [38]. We used the HSDQ for our complete sample, since the HSDQ has also been previously used in various adolescent Dutch samples [39,40,41,42,43,44,45,46,47] and broad screening measures for sleep disorders developed for adolescents aged 12 years and above are scarce [48,49]. It consists of 32 items in which sleep symptoms are rated over the past three months on a five-point scale (1 = “not at all applicable” to 5 = “applicable”). The scale is divided into six subscales measuring the ICSD-2 diagnostic domains: (1) Insomnias: 8 items, (2) CRSD: 6 items, (3) SBD: 4 items, (4) Hypersomnia: 5 items, (5) Parasomnias: 6 items, and (6) RLS/LMS: 5 items. Each subscale uses cut-off scores which are calculated by dividing the total score by the number of items of the specific diagnostic subdomain, resulting in a dichotomous outcome (yes/no disorder). Sensitivity ranges per diagnostic domain from 87–99 and specificity from 90–98 (only lower for insomnia 0.69). The HSDQ showed satisfactory internal consistency (α = 0.90), overall accuracy (88%) and substantial-to-excellent agreement (kappa = 0.80) with the primary diagnoses in a sleep clinic and has already been used in a large epidemiological Dutch cohort [50].

#### 2.3.2. Sociodemographic, Cancer-Related Factors and the Presence of Comorbid Health Conditions

Participants completed an additional self-report questionnaire including sociodemographic variables (age, sex, country of birth, educational level, living situation). Age was additionally categorized in adolescents (aged 12–17 years old) and young adults (18–26 years old). Educational level was categorized as low–middle and high according to the Dutch standard education classification [51]. Participants also completed self-report items on cancer-related factors: their type of cancer diagnosis, received cancer treatment, age at diagnosis. In addition, they indicated if any co-morbid health conditions (either medical or psychological) were present (self-reported, open-ended question), which we categorized dichotomously (yes/not present). For non-responders, age at study invitation, sex, age at diagnosis, time since diagnosis and type of cancer diagnosis were verified in the DCOG database.

#### 2.3.3. Sleep Treatment Needs

Furthermore, questions were included regarding whether participants currently received sleep treatment, whether they received treatment in the past and if they wanted sleep treatment at the moment. In addition, participants were asked to indicate sleep medication use in the past month (either prescribed or over-the-counter), specified as either benzodiazepine, melatonin or homeopathic remedies.

### 2.4. Statistical Analyses

For all analyses, IBM SPSS version 25 was used. Characteristics for responders and non-responders were compared using chi-squared tests. Prevalence rates of sleep disorders and treatment needs were evaluated with descriptive statistics and for young adults descriptively compared with data from Kerkhof (2017) [50] including adults from the general population. Logistic regression analyses were used to analyze risk factors (cancer-related factors, sex, age group and co-morbid health conditions) for sleep disorders. A minimum number of events of 10 was set to ensure enough power in the analyses [52], which was only the case for the outcomes: insomnia, CRSD and RLS/LMS. Missing data on the outcome (HSDQ) was imputed using the personal means on the diagnostic subdomain if ≤10% was missing in the individual case. If more than 10% was missing in the individual case, the response was excluded using listwise deletion.

## 3. Results

### 3.1. Sample Characteristics

In total, 1032 patients were invited, of whom 576 participated (response rate 55.8%). See Appendix A for a flow chart of patient inclusion and see Table 1 for a detailed description of the sample characteristics. Responders differed significantly from non-responders in sex; females were more likely to respond (χ^2^ = 8.18, *p* < 0.01). All other variables, e.g., type of cancer diagnosis, did not differ (see Table 1).

### 3.2. Prevalence of Sleep Disorders

The prevalence rates are shown in Table 2. The most commonly subjective sleep disorder was insomnia (9.6%), followed by CRSD (8.1%) and RLS/LMS (7.6%). SBD was the least prevalent (1.8%). In young adults, insomnia (12.4% in our sample vs. 8.2% in the general adult population) and CRSD (11.6% in our sample vs. 5.3% in the general adult population) were more prevalent compared to adults from the general population. Other disorders seem to be similar or less prevalent. In total, 97 patients (17.2%) reported one or more sleep disorders. Sleep disorders often co-occurred; about half of the respondents (*n* = 52) had two or more sleep disorders. The most common co-morbidity was insomnia and CRSD (*n* = 30).

### 3.3. Risk Factors

#### 3.3.1. Sex

Female sex was a risk factor for insomnia and RLS/LMS, but not for CRSD (*p* = 0.13) in univariate analyses. When analyzed in a multivariable model, female sex remained a risk factor for insomnia (OR 2.33, 95% CI 1.25–4.35), but not for RLS/LMS (*p* = 0.06); see Table 3.

#### 3.3.2. Age Group: Adolescents Versus Young Adults

Young adults were more at risk than adolescents for insomnia and CRSD, but not for RLS/LMS (*p* = 0.25) in univariate analyses. When analyzed in a multivariable model, the effect was not sustained in insomnia (*p* = 0.15) but remained a risk factor in CRSD (OR 2.15, 95% CI 1.13–4.07); see Table 3.

#### 3.3.3. Co-Morbid Health Conditions

Patients with a co-morbid health condition had an increased risk for insomnia, CRSD and RLS/LMS in univariate analyses. When analyzed in a multivariable model with sex and/or age group, the effect was sustained in all (OR 2.89–3.02); see Table 3.

#### 3.3.4. Childhood Cancer-Related Risk Factors

Age at diagnosis (in years), cancer diagnosis, type of oncological treatment and time after treatment (in years) were not significant risk factors (*p* > 0.05) for insomnia, CRSD and RLS/LMS in univariate analyses; see Table 3.

### 3.4. Sleep Medication Use

Of all participants, 11.2% used sleep medication (*n* = 64). Melatonin was the most commonly used type of sleep medication (8.9%; *n* = 51) followed by homeopathic remedies (1.4%, *n* = 8) and benzodiazepines (1.1%, *n* = 6). See Appendix A for an overview of sleep medication use per sleep disorder and for those without a sleep disorder. Half of the patients that used melatonin did not meet the HSDQ criteria for a sleep disorder at the time of the study.

### 3.5. Treatment Needs

Of those with a sleep disorder, 42–72%, depending on the type of sleep disorder, indicated they would like help to sleep better, with the highest percentage in the insomnia group; see Figure 1. Nevertheless, only 0–5.6% of patients with a sleep disorder were currently receiving some form of sleep treatment. About a quarter (20.0–27.8%) had received sleep care in the past. Of those without a sleep disorder, 9.9% wanted help to sleep better.

## 4. Discussion

This national cohort study described sleep disorders in adolescent and young adult childhood cancer patients during follow-up. In a large sample representing all types of childhood cancer diagnoses, we found that the prevalence of six subjective sleep disorders ranged from 1.8–9.6%. Insomnia and circadian rhythm sleep disorders were most commonly reported and were more prevalent than in the general population. The majority of childhood cancer patients at follow-up with a sleep disorder expressed a need for sleep treatment, but only a small minority actually received sleep treatment at that time, indicating an unmet need.

Most studies in the general population estimate the prevalence of insomnia diagnosis in adults between 6–10%, which differs depending on the tool or diagnosis method used to determine insomnia [53,54]. Studies in young adults and/or adolescents are more sparse than adult studies. Using the same instrument as in our study, the HDSQ, the rate of insomnia in adults (18–70 years old) from the Dutch general population was found to be lower (8.2%) [50] compared to our young adults (12.4%). Even though this study also included older adults, the authors reported no effect of age within their population. Therefore, the difference in insomnia prevalence between both studies does not seem to be associated with age in adults. Reported prevalence rates of insomnia in young adults in a similar age range (20–29 years) from the general population vary between 8.9% [5] and 14.9% (39), although other assessment tools were used. In adolescents from the general population, no previous studies using the HSDQ have been reported. Using other tools to assess insomnia, rates of 4–9.4% have been found [55,56,57] compared to 7.4% in the adolescents who participated in our study, which is comparable.

CRSD is a sleep disorder often occurring in young adults, since changes in circadian rhythm are part of a developmental milestone [58]. Our young adults also reported a higher rate of 11.6% compared to 5.6% in the adolescents. The prevalence was also higher compared to the general adult population (5.3%, 18–70 years old, using the HSDQ [50]), independent of age differences. CRSD rates follow the same pattern as the insomnia rate with regard to the COVID-19 pandemic, with higher levels in non-pandemic circumstances. In the literature, prevalence rates of CRSD are typically evaluated using its most common type: delayed sleep phase disorder (DSPD). In a study focusing on young adults, 4% reported DSPD [59]. In adolescents, rates of 1.1% DSPD [60] and 0.4% CRSD [57] compared to 5.6% in our sample have been reported. Although there are differences in its definition, it seems that CRSD might be increased in childhood cancer patients during follow-up, even more in non-pandemic circumstances. An explanation for the higher prevalence of CRSD in childhood cancer patients might lie in its relationship with fatigue. Fatigue is a common side effect of cancer treatment and tends to persist in childhood cancer survivors who report increased levels of fatigue [61]. A disrupted circadian rhythm is associated with fatigue [21,62], and inconsistent sleep habits during treatment [63] might persist after treatment has ended.

The other four sleep disorders: RLS/LMS, parasomnia, hypersomnia and SBD seem comparable or less common in childhood cancer survivors compared to the general adult population [50]. Compared to studies in young adults [6,64] and adolescents from the general population [65] using different tools, RLS/LMS seems slightly more common. Studies on hypersomnia and parasomnia in adolescents/young adults are sparse; mostly, excessive daytime sleeping or aspects of parasomnia, e.g., sleep walking, are described, which are not comparable to our diagnosis-based outcome. SBD appears to be less prevalent compared to studies using polysomnographic methods in young adults [6] and adolescents [66], but this difference might also lie in the different measurement methods used. Overall, sleep disorders negatively impact health and require specialized treatment [12,26], so, regardless of their prevalence, it remains important to differentiate these according to reported sleep symptoms in clinical practice.

Risk factors that emerged from our findings were: female sex, young adult age-group and co-morbid health conditions, which is in line with what is known from the general population [4,5,6,11,50], although causal inferences cannot be made from this study. While some co-morbid conditions may lead to poor sleep, some studies suggest the opposite. In a longitudinal study including children with sleep problems, they were more likely to develop new medical disorders at follow-up [1]. It might also be that sleep disorders cluster with other problems, such as fatigue, depressive symptoms or cognitive deficits, which together interrelate with poorer health outcomes [14,67].

We did not find differences between type of cancer diagnosis, oncological treatment or age of diagnosis and time after treatment. A previous childhood cancer survivor study (CCSS) also did not demonstrate a difference between types of diagnosis or treatment and sleep [30]. However, we did expect a difference between CNS tumors and other types of cancer diagnoses due to the potential influence of tumor localization on sleep [26], although previous findings seem mixed [31,35,68,69,70]. Some studies suggest more sleep problems in those with a tumor in the suprasellar region [24,26,71,72]. Our sample only included a very small group of patients who may be at increased risk within the CNS category (e.g., only seven with craniopharyngioma) and we also included other CNS diagnoses that may not be at more risk for sleep disorders than other cancer diagnoses, such as astrocytoma, at a location unrelated to sleep regulation.

Sleep medication was used by about one in ten patients, melatonin for the most part. This melatonin usage rate of about 9% is higher than in the Dutch general population: 6% in children [73] and 3.5% in adults [50]. Melatonin is an over-the-counter medication available in the Netherlands. It is recommended as a short-term treatment for CRSD, problems with sleep-onset or jet lag [74]. It might be that the potential increased rate of CRSD in childhood cancer survivors explains the more common use of melatonin in this group, but inappropriate timing of melatonin can also cause CRSD [75].

A need for sleep treatment was reported by a majority of childhood cancer patients with a sleep disorder, but only a few actually received treatment. In previous research using qualitative methods, adolescents and young adults also indicated that at the end of cancer treatment there is a need for more information and sources of support for common psychosocial issues, such as pain, fatigue and insomnia [76]. Additionally, childhood cancer survivors indicated an unmet need and the difficulty they have finding fitting psychosocial care [77]. Our findings also confirm this for sleep disorders, which is comparable to low access to sleep care in the general population [10,36], and for adults who had cancer [78]. Helping childhood cancer survivors to find appropriate sleep treatment can fulfill this unmet need and improve their quality of life [79].

### Strengths and Limitations

Our findings should be interpreted in the light of several strengths and limitations. A strength of this study is that we included a large patient population with diverse cancer diagnoses which are representative for the Dutch population based on the Dutch Childhood Oncology Group (DCOG) database. However, several limitations should be taken into account. First, although we used a validated instrument which measures sleep disorders reliably according to the ICSD-2, it was only validated in adults (18+) and not in adolescents (12–17 years old). With regard to content validity, adolescents might interpret items differently to adults. Second, we also could not include reference norm data to statistically compare the results. Third, we used a self-report measure and not a clinical interview or polysomnography, which are the gold standards to diagnose sleep disorders [19]. However, a self-report measure is more feasible and less burdensome for screening. Fourth, we measured partly pre-pandemic and partly during the COVID-19 pandemic, which may have influenced our results. Van Gorp and colleagues (2021) [80] found a decrease in sleep problems in childhood cancer patients during the pandemic regulations due to being homeschooled in some phases of the pandemic and therefore waking up less early. In additional post hoc analyses, we also found a decrease (see Appendix A). The total rates might therefore be an underestimation; however, some studies report an increase of sleep problems during the pandemic [81,82]. Finally, we found in our non-responder analyses that females were more likely to respond, which could constitute a response bias. Females also report more sleep problems [6], so might therefore be more likely to participate, potentially leading to overestimation.

## 5. Conclusions

Insomnia and CRSD seem to be prevalent in adolescents and young adult childhood cancer patients in follow-up and in survivorship. Young adults seem specifically at risk and the majority report an unmet sleep treatment need. Future research should focus on a more detailed inspection of sleep disorders in childhood cancer patients during follow-up compared to age-matched norms and explore how treatment needs can be met. In clinical practice, systematic monitoring of sleep after cancer treatment can contribute to increased recognition of sleep disorders, supporting survivors to find appropriate sleep treatment and thereby help adolescent and young adult childhood cancer patients after completion of treatment to maintain optimal health and quality of life.

## Figures and Tables

**Figure 1 cancers-14-00926-f001:**
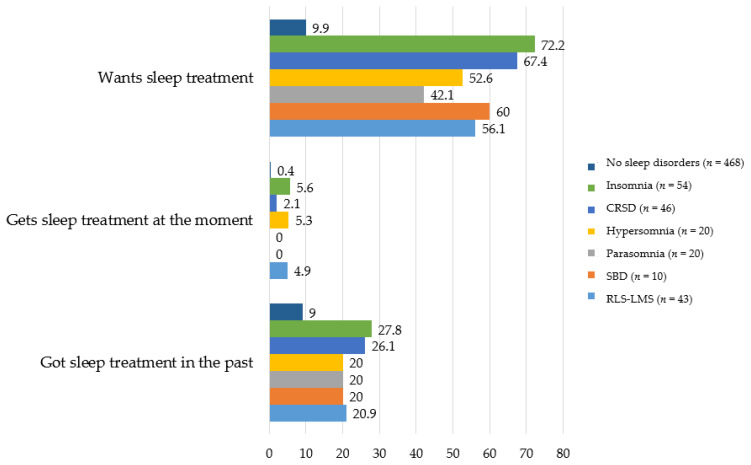
Treatment needs and use per sleep disorder in percentages.

**Table 1 cancers-14-00926-t001:** Sample characteristics.

Characteristics	Responders (*n* = 576)	Non-responders (*n* = 456)	*p*-Value
*Sociodemographic*			
Age at study invitation (y), mean (SD)	17.00 (2.91)	17.59 (2.91)	0.18
Age group, *n* (%)			
Adolescents (12–17 y)	328 (56.9)	260 (57.0)	0.98
Young adult (18–26 y)	248 (43.1)	196 (43.0)	
Sex, *n* (%)			
Female	285 (49.5)	184 (40.4)	<0.01
Current educational level, *n* (%)			
Low	139 (24.1)	NA	
Middle	326 (57.1)		
High	104 (18.1)		
Country of birth, *n* (%)			
The Netherlands	548 (95.1)	NA	
Other	20 (3.5)		
*Childhood cancer-specific*			
Age at diagnosis (y), mean (SD)	12.83 (3.19)	13.52 (2.62)	0.89
Time since diagnosis in years, mean (SD)	3.95 (2.35)	3.81 (1.98)	0.28
Diagnosis groups, *n* (%)			
Hemato-oncology	256 (44.4)	212 (46.6)	
Neuro-oncology	136 (23.6)	95 (20.9)	0.52
Solid	184 (31.9)	148 (32.5)	
Time since end of treatment in years, mean (SD)	3.21 (2.23)	NA	
Type of oncological treatment, *n* (%)			
No treatment	8 (1.4)	NA	
Chemotherapy	423 (73.4)		
Radiation	147 (25.5)		
Surgery	312 (54.2)		
Stem cell transplantation	36 (6.3)		
Mixed/Other	29 (5.0)		
Comorbid health problems, *n* (%)			
No	462 (80.2)	NA	
Yes	102 (17.7)		
Medical	68 (11.8)		
Psychological	34 (5.9)		

Note. NA: information not available for non-responders. Subheadings are presented in italics.

**Table 2 cancers-14-00926-t002:** Prevalence rates of subjective sleep disorders in the total group, per sex and per age group, with comparison to published data.

Sleep Disorders	Childhood Cancer Cohort					Published Data ^1^
	Total Group *N* (%) *n* = 565	Males, *n* (%) *n* = 282	Females, *n* (%) *n* = 282	Adolescents 12–17 y, *n* (%) *n* = 323	Young Adults 18–26 y, *n* (%) *n* = 242	Adults 18–70 y from the General Population ^1^, *n* (%) *n* = 2089
Insomnia	54 (9.6)	16 (5.7)	38 (13.4)	24 (7.4)	30 (12.4)	171 (8.2)
CRSD	46 (8.1)	18 (6.4)	28 (9.9)	18 (5.6)	28 (11.6)	110 (5.3)
RLS/LMS	43 (7.6)	15 (5.3)	28 (9.9)	21 (6.5)	22 (9.1)	261 (12.5)
Parasomnia	20 (3.5)	3 (1.1)	17 (6.0)	9 (2.8)	11 (4.5)	128 (6.1)
Hypersomnia	20 (3.5)	4 (1.4)	16 (5.7)	6 (1.9)	14 (5.8)	124 (5.9)
SBD	10 (1.8)	2 (0.7)	8 (2.8)	3 (0.9)	7 (2.9)	148 (7.1)

Note. ^1^ Data shown in article Kerkhof (2017).

**Table 3 cancers-14-00926-t003:** Risk factors for sleep disorders according to the HSDQ (OR and 95% confidence interval).

Risk Factors	Insomnia (*n* = 54)		CRSD (*n* = 46)		RLS/LMS (*n* = 43)	
	Univariate	Multivariable	Univariate	Multivariable	Univariate	Multivariable
Female	2.59 (1.41–4.76) **	2.33 (1.25–4.35) **	1.62 (0.87–3.00)	-	1.96 (1.02–3.76) *	1.89 (0.98–3.66)
Age group: young adults (ref: adolescents)	1.76 (1.00–3.10) *	1.55 (0.86–2.80)	2.22 (1.20–4.11) *	2.15 (1.13–4.07) *	1.44 (0.77–2.68)	-
Having a comorbid health condition	3.00 (1.62–5.53) ***	2.89 (1.55–5.40) **	3.14 (1.64–6.02) **	2.98 (1.55–5.74) **	3.04 (1.57–5.91) **	3.02 (1.55–5.89) **
Age at diagnosis in years	1.04 (0.95–1.14)	-	1.04 (0.94–1.16)	-	1.08 (0.97–1.21)	-
Cancer diagnosis: (ref: CNS)						
Hemato	0.94 (0.46–1.92)	-	2.15 (0.86–5.43)	-	1.02 (0.46–2.25)	-
Solid	1.01 (0.48–2.13)	-	2.17 (0.83–5.66)	-	1.02 (0.44–2.37)	-
Type of oncological treatment ^1^: (ref: Radiation and/or SCT)						
Surgery only	1.07 (0.74–1.54)	-	0.84 (0.56–1.26)	-	0.89 (0.59–1.36)	-
Chemotherapy with or without surgery	0.92 (0.64–1.32)	-	1.18 (0.78–1.77)	-	1.09 (0.72–1.66)	-
Time since end of treatment (in years)	1.05 (0.93–1.18)	-	1.07 (0.94–1.21)	-	1.03 (0.90–1.18)	-

Note. OR and 95% CI are presented; if significant, asterisks were added: * <0.05, ** <0.01, *** <0.001. Risk factors for parasomnia, hypersomnia and SBD could not be analyzed due to small sample sizes (≤10 per event). ^1^ Categorized on treatment intensity. The radiation and/or SCT group is the most intensively treated group; they could also have had chemotherapy and/or surgery.

## Data Availability

The data presented in this study are available on request from the corresponding author. The data are not publicly available due to privacy restrictions.

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
