# Peer review of "Prevalence of Sleep Disorders, Risk Factors and Sleep Treatment Needs of Adolescents and Young Adult Childhood Cancer Patients in Follow-Up after Treatment"

_cancers, 2022, doi:10.3390/cancers14040926_

Round 1
Reviewer 1 Report
The authors reported results from a study investigating sleep disorders according to the ICSD-2 in a cohort of childhood cancers patients.
Few studies have been published in the field about sleep difficulties in cancer patients and particularly in childhood cancers, this study bring thus new knowledge related to this topic. Limitations of the study are clearly described and aknowledged which is really appreciated. I have several suggestions to improve the manuscript and particulalry the abstract. Also, inclusion criteria are not clear enough.
1- Simple summary
- Even if I see the point of being one of the first study doing such investigation, I would emove this term in the abstract and in the manuscript as it is always difficult to definitively state this.
- Be clear than this study is related to subjective sleep difficulties instead of using "self-reported" in parentheses.
- Add that the questionnaire used in the study was a validated sleep questionnaire. Most previous studies used only a question or an item from a questionnaire dedicated to quality of life and this will highlight the strenght of the study.
- Precise that sleep disorders were classified according to the ICSD-2, if possible with words limit.
- Precise that melatonin was reported as the most sleep treatments among others. Instead, it is not clear why authors talk only about melatonin.
2- Abstract
- In the first sentence, please specify "subjective complaints".
- The sentence "Descriptive statistics..." is not clear enough, please improve it.
- In the Results, please indicate "response rate 55.8%".
- CRSD was not define before in the abstract.
- Conclusion: please specify that it concerns childhood cancers compared to previsouly published data.
Keywords: I miss the meaning of "AYA". Please specify.
3- Introduction
- Line 59-60 "In a longitudinal study...": in which population ? Healthy volunteers out of cancer pathology ? Please specify. Also, specify compare to what the medical disorder is new.
- Lines 92-93 "For example during ALL...": The sentence is unclear, please explain how circadian sleep activity rhtyhm can modify CRSD.
- In the aims, please specify that this concern childhood cancers.
4- Methods
- What is LATER ?
- Why only participants having finished their treatment since at least 6 months was an inclusion criteria ? This criteria should be added as a limit.
- What is the meaning of "physician specific reasons"?
- Although this is stated in the limits, authors should clearly say than the HSDQ is validated only in adults and give a reason why this questionnaire was used instead of another. Other questionnaires have been validated for young adults and adolescents (See for example Sen and Spruyt, 2020 as a review).
- In the treatments report, I am surprise that none of participants received CBT. Was this proposed in the answers ?
5- Results
- Figure 1 could be improve for example using colors or by filling histograms with stripes for example.
- I am not sure that results related to the pandemic should be kept in the article as it seems out of the first idea of the study. I thus suggest to remove this as well as the related discussion.
- Other results are clearly described.
6- Discussion
- Lines 10-11, "Most studies estimate...": please specify the population considered.
- The discussion related to pandemic results seems out of the first idea of the study and should be more discussed. Most of previous studies reported more sleep prevalence during the pandemic period. Again, I suggest to remove these results.
- The limitations and the conclusion are very clear and appropriate.
Author Response
Please see the attachment for the response letter.

Reviewer 2 Report
This study reports on the prevalence of sleep disorders, associated risk factors, and perceived need for sleep treatment in adolescent and young adult survivors of childhood cancer. Overall, this is an important topic that is understudied in the adolescent and young adult cancer population.
The rationale for the study is presented clearly, and the methodology appears appropriate to the aims of the study.
The major flaw of this study is the lack of validation of the Holland Sleep Disorders Questionnaire in the adolescent population. The authors note this as a limitation in the discussion section, though I believe more clarity should be added as to the rationale for choosing this questionnaire for the 12-17-year-olds in their study. Has this questionnaire been used in a pediatric population in the past, or is this the first study to do so? I believe it will add transparency to the methodology to make clear the rationale for utilizing a measure that is not validated for the population on which it is used. This is especially important since the 12-17 year old age group was well over half the overall sample.
The study is otherwise well written and clearly presented.
Author Response

(The authors gave the same response as above.)
